# Targeting Ferroptosis: Emerging Insights into Osteoporosis Mechanisms

**DOI:** 10.3390/biology14081062

**Published:** 2025-08-15

**Authors:** Hailing Yang, Kang Ru, Shuai Liu, Chunyu Zhu, Airong Qian, Zhihao Chen

**Affiliations:** 1Laboratory for Bone Metabolism, Xi’an Key Laboratory of Special Medicine and Health Engineering, Key Laboratory for Space Biosciences and Biotechnology, Research Center for Special Medicine and Health Systems Engineering, School of Life Sciences, Northwestern Polytechnical University, Xi’an 710129, China; yanghailing@mail.nwpu.edu.cn (H.Y.); rukang@mail.nwpu.edu.cn (K.R.); zcy@nwpu.edu.cn (C.Z.); 2Department of Obstetrics and Gynecology, Xijing Hospital, The Fourth Military Medical University, Xi’an 710032, China; liushuaiwz@126.com

**Keywords:** ferroptosis, osteoporosis, iron overload, osteoblasts, osteoclasts

## Abstract

**Simple Summary:**

Osteoporosis is a common condition caused by an imbalance in bone remodeling, where bone breakdown surpasses bone formation, leading to weakened bones and a higher risk of fractures. Recent research has identified ferroptosis—a form of cell death driven by iron overload and lipid damage—as a key contributor to this imbalance. Ferroptosis impairs the development and function of osteoblasts, the cells that build bone, while enhancing the activity of osteoclasts, which break down bone. This dual effect reduces bone formation and increases bone loss, ultimately disrupting bone health. Understanding the role of ferroptosis in osteoporosis may support the development of new treatments that help restore bone balance and reduce the global burden of this disease.

**Abstract:**

Ferroptosis, a distinct form of programmed cell death characterized by iron-dependent lipid peroxidation, has emerged as a critical factor in the pathogenesis of various diseases. Given the increasing prevalence of osteoporosis worldwide and the increasing incidence of osteoporosis, understanding the molecular mechanisms underlying bone loss is imperative for developing targeted therapies. Recent evidence suggests that ferroptosis plays a pivotal role in osteoporosis by influencing the balance between osteoblast and osteoclast activity. This review examines the mechanistic basis of ferroptosis and its pathological implications in osteoporosis. By delineating the interplay between ferroptosis and skeletal remodeling, we highlight potential therapeutic strategies aimed at modulating ferroptosis to mitigate osteoporosis progression.

## 1. Introduction

Ferroptosis, a recently identified form of programmed cell death, is characterized by uncontrolled iron accumulation and lipid peroxidation [1,2]. The term “ferroptosis” was first introduced by Dixon et al. in 2012, following their discovery that erastin—a small molecule selectively lethal to oncogenic RAS-mutant cells—induces a distinct form of iron-dependent, non-apoptotic cell death [3]. Notably, the groundwork for this discovery was laid as early as 2003, when Dolma et al. reported that compounds like erastin and RSL3 triggered death in RAS-mutant tumor cells through a mechanism distinct from classical apoptosis [4]. Over the past decade, growing evidence has linked ferroptosis to various diseases, including cancer, neurodegeneration (e.g., Parkinson’s disease), cardiovascular conditions (e.g., heart failure and atherosclerosis), liver disorders (e.g., non-alcoholic fatty liver disease), viral infections, osteoporosis, and ischemia–reperfusion injury, among others [5,6,7,8]. As a result, ferroptosis modulation is now viewed as a promising therapeutic strategy across a broad spectrum of diseases.

Osteoporosis is a systemic skeletal disorder characterized by reduced bone mineral density (BMD) and deterioration of bone microarchitecture, resulting in an increased risk of fragility fractures [9,10]. Its prevalence increases significantly with age, making it a major public health concern among the elderly. As of March 2020, global epidemiological data indicated a prevalence rate of 21.7% among older adults, with postmenopausal women disproportionately affected [11]. Skeletal tissue is a dynamic metabolic structure whose integrity relies on the maintenance of bone homeostasis. Central to this process is bone remodeling—a tightly regulated mechanism involving the coordinated actions of osteoclasts, which resorb old bone, and osteoblasts, which synthesize new bone [12,13,14]. This interplay ensures the continual renewal and repair of the skeletal architecture. Osteoporosis arises from disruption of this delicate balance in bone remodeling. Mechanistically, this imbalance may be attributed to iron overload. On one hand, excess iron can impair the differentiation of osteoblasts, leading to reduced bone formation. On the other, it may induce hyperactivation of osteoclasts, resulting in excessive bone resorption that surpasses new bone synthesis [15,16,17].

In recent years, accumulating evidence has highlighted a critical association between ferroptosis and the development of osteoporosis [5,6,18]. Liu et al. initially reviewed this connection and proposed a conceptual framework for understanding how ferroptosis may contribute to bone loss [19]. More recent work has expanded upon this foundation by exploring various factors that influence ferroptosis in bone metabolism. Among these, the roles of immune regulation, iron homeostasis, and oxidative stress have emerged as important mediators of bone cell function [20]. Building upon this, the present review provides a more comprehensive and in-depth analysis of ferroptosis mechanisms across distinct bone cell types (e.g., osteoblasts and osteoclasts), with particular emphasis on molecular pathways and regulatory networks. Furthermore, we offer a systematic overview of emerging therapeutic strategies, including traditional iron chelators and novel natural antioxidants. We believe this review will serve as a valuable reference for advancing both basic and translational research in this evolving field.

## 2. Overview of Ferroptosis

The term ferroptosis refers to a regulated form of programmed cell death characterized by the accumulation of membrane-damaged, iron-dependent lipid hydroperoxides to lethal levels [21]. Alongside ferroptosis, other forms of programmed cell death include apoptosis, necrosis, pyroptosis, and autophagy [22]. Apoptosis, typically non-inflammatory, involves characteristic morphological and biochemical changes, such as chromatin condensation, apoptotic body formation, DNA fragmentation, caspase activation, and phosphatidylserine externalization [23,24,25,26]. In contrast, necrosis is characterized by cell swelling, loss of membrane integrity, and uncontrolled release of intracellular contents, often triggering an inflammatory response [27]. Pyroptosis, an inflammatory form of programmed cell death, is induced by infection or cellular damage, leading to the activation of gasdermin proteins, cell swelling, membrane pore formation, and the release of proinflammatory cytokines [28]. During autophagy, autophagosomes engulf cytoplasmic contents and deliver them to lysosomes for degradation and recycling [29]. While ferroptosis shares certain regulatory features with other death pathways, it remains distinct in its morphological, biochemical, and genetic characteristics [3,30,31]. Ferroptosis is notably different from the other death forms. Morphologically, ferroptosis is characterized by mitochondrial shrinkage, reduced mitochondrial number, increased membrane density, loss or simplification of cristae, preservation of nuclear integrity, and the formation of membrane-bound bubbles without rupture [3,32,33]. From a biochemical perspective, the main hallmarks of ferroptosis include elevated levels of unstable iron, accumulation of phospholipid hydroperoxides, and inactivation of glutathione peroxidase 4 (GPX4) [34]. From a genetic standpoint, ferroptosis is regulated by a complex network of genes and signaling pathways, which encompass genes related to iron metabolism (e.g., transferrin receptor (TFRC), ferritin heavy chain (FTH1), and ferritin light chain (FTL)), lipid metabolism (e.g., acyl-CoA synthetase long-chain family member 4 (ACSL4) and members of the arachidonate lipoxygenase (ALOX) family), and the antioxidant defense system (e.g., GPX4 and solute carrier family 7 member 11 (SLC7A11)). Additionally, key regulatory proteins such as p53 and nuclear factor erythroid 2-related factor 2 (Nrf2), along with signaling pathways including the Kelch-like ECH-associated protein 1–Nrf2 (KEAP1–Nrf2) axis and the mechanistic target of rapamycin (mTOR) pathway, are involved in the regulation of ferroptosis [35,36,37,38,39,40,41,42]. Collectively, these factors orchestrate the regulation of cellular iron homeostasis, lipid metabolism, and antioxidant defenses, which are crucial for the execution of ferroptosis.

### 2.1. Iron Accumulation and Dysregulation in Ferroptosis

Iron is an essential trace element involved in numerous physiological processes in the human body [43]. One of the defining features of ferroptosis is the dysregulated accumulation of iron, which leads to disturbances in iron metabolism. Intracellular iron is primarily stored in the forms of Fe^3+^ and Fe^2+^. Fe^3+^ is the predominant form of iron in transit within the body and enters cells by binding to transferrin receptor 1 (TFR1) on the cell membrane. Once inside the cell, Fe^3+^ is reduced to Fe^2+^ by six-transmembrane epithelial antigen of prostate 3 (STEAP3) [44]. The resulting Fe^2+^ is transported into the labile iron pool (LIP) by divalent metal transporter 1 (DMT1). Excess Fe^2+^ is exported from the cell via ferroportin 1 (FPN1) and stored in the form of ferritin, primarily as ferritin heavy chain 1 (FTH1) and ferritin light chain 1 (FTL1). Under normal physiological conditions, ferritin maintains intracellular iron in an inert form, preventing lipid peroxidation. However, ferritinophagy—a process mediated by nuclear receptor coactivator 4 (NCOA4)—promotes the degradation of ferritin and the release of Fe^2+^. This increases the labile iron pool and contributes to intracellular iron overload [45]. This iron overload can, in turn, initiate ferroptosis through the generation of reactive oxygen species (ROS) via the Fenton reaction [46].

### 2.2. Lipid Peroxidation and Polyunsaturated Fatty Acids (PUFAs)

Lipid peroxidation is another hallmark of ferroptosis, with abnormalities in lipid metabolism playing a critical role in its initiation. PUFAs, such as arachidonic acid (AA) and adrenic acid (AdA), are highly prone to oxidation by ROS, resulting in lipid peroxidation [47]. Hydroxyl radicals can directly interact with PUFAs in membrane phospholipids through chain reactions, forming lipid peroxides. These lipid peroxides subsequently attack the cell membrane, inducing the morphological changes associated with ferroptosis [48]. Acyl-CoA synthetase long-chain family member 4 (ACSL4) and lysophosphatidylcholine acyltransferase 3 (LPCAT3) incorporate PUFAs into cellular phospholipids, particularly phosphatidylethanolamine (PE), to form PUFA-PE. This PUFA-PE is then further oxidized into lipid peroxides by lipoxygenase (LOX) or cytochrome P450 oxidoreductase (POR), thereby driving ferroptosis [48].

### 2.3. The System Xc^−^–GPX4 Axis in Antioxidant Defense

System Xc^−^, a heterodimeric glutamate/cystine transporter located on the cell surface, plays a pivotal role in maintaining intracellular redox homeostasis. Its primary function is to mediate the exchange of intracellular glutamate and extracellular cystine. Once transported into the cell, Cys^2+^ is rapidly reduced to cysteine, which serves as a rate-limiting precursor for the synthesis of glutathione (GSH) via the enzymatic actions of glutamate–cysteine ligase (GCL) and glutathione synthetase (GSS). GPX4 utilizes reduced GSH as a substrate to detoxify lipid hydroperoxides (L-OOH) into non-toxic lipid alcohols (L-OH) and to convert hydrogen peroxide (H_2_O_2_) into water [3,32,38,49]. Inhibition of system Xc^−^ leads to a decrease in intracellular GSH and a reduction in GPX4 activity, which exacerbates ROS accumulation, ultimately triggering ferroptosis [50,51]. Therefore, the system Xc^−^ antioxidant system plays a crucial role in the occurrence of ferroptosis.

### 2.4. Regulation of Ferroptosis by p53

The tumor suppressor p53 not only governs cell cycle arrest, apoptosis, and DNA repair but also exerts significant regulatory control over ferroptosis [52,53]. p53 downregulates the expression of the system Xc^−^ light-chain subunit SLC7A11, thereby inhibiting cystine transport, reducing cystine-dependent GPX activity, and diminishing the cellular antioxidant capacity. This imbalance increases the generation of lipid ROS, ultimately driving ferroptosis [39]. Under conditions of ROS-induced stress, p53 can further activate spermidine/spermine N1-acetyltransferase 1 (SAT1)-mediated polyamine metabolism, thereby enhancing lipid peroxidation and ferroptosis [54]. In addition, p53 modulates ferroptosis through noncoding RNAs. For example, the lncRNA PVT1 has been shown to induce ferroptosis by suppressing miR-214-mediated downregulation of TFR1 [55]. Conversely, p53 can also function as a negative regulator of ferroptosis, notably by upregulating target genes such as *CDKN1A* and *DPP4* to suppress ferroptotic processes [56]. Collectively, these findings indicate that p53 plays a dual role in ferroptosis, acting as either a promoter or inhibitor depending on cell type, environmental context, and its activation status. The regulation of ferroptosis by p53 is, therefore, a complex, multifaceted, and context-dependent process involving diverse molecular mechanisms and signaling pathways.

### 2.5. Summary of Ferroptosis Mechanisms

In summary, ferroptosis is driven by iron-mediated ROS generation and lipid peroxidation, which disrupt membrane integrity, coupled with the failure of antioxidant defenses. This distinct mechanism sets ferroptosis apart from other forms of cell death, such as apoptosis, necrosis, and autophagy. A schematic overview of these mechanisms is provided in Figure 1.

Ferroptosis is an iron-dependent form of regulated cell death caused by lipid peroxidation. TFR1 mediates iron uptake, where Fe^3+^ is reduced by STEAP3 and transported as Fe^2+^ into the LIP via DMT1 or FPN1. Excess Fe^2+^ promotes ROS generation, triggering lipid peroxidation. PUFAs are converted into PUFA-PEs by ACSL4 and LPCAT3, which are then oxidized by LOX/POR, leading to ferroptosis. The antioxidant enzyme GPX4 reduces L-OOH via GSH, which is synthesized from cystine via GCL and the glutathione synthetase GSS. GPX4 inhibition or GSH depletion disrupts this defense, promoting ferroptosis. p53 downregulates the expression of SLC7A11, the light-chain subunit of system Xc^−^, thereby inhibiting cystine uptake, reducing glutathione synthesis, and impairing GPX4 activity. Under oxidative stress conditions, p53 can also activate SAT1-mediated polyamine metabolism to enhance lipid peroxidation, as well as induce lncRNA PVT1 expression, which promotes TFR1 upregulation and iron accumulation. These pathways collectively contribute to p53-mediated promotion of ferroptosis.

Arrow colors: Blue, iron metabolism and dysregulation; orange, lipid peroxidation and PUFA metabolism; black, system Xc^−^–GPX4 axis; purple, regulation by p53 and its downstream effectors (SAT1, lncRNA PVT1).

## 3. Pathogenesis of Osteoporosis

Osteoporosis is characterized by an imbalance in the bone remodeling process in which bone resorption exceeds bone formation [57]. Bone structure and strength are maintained by the dynamic balance between osteoclast-mediated bone resorption and osteoblast-mediated bone formation. Dysfunction in either osteoclasts or osteoblasts disrupts this balance, leading to reduced bone density and increased bone fragility. The pathogenesis of osteoporosis is multifactorial and involves a combination of genetic predispositions, hormonal imbalances—particularly estrogen deficiency—vitamin D deficiency, and inadequate calcium intake, all of which contribute significantly to disease progression [58,59,60,61]. Additionally, inflammatory mediators and oxidative stress have been identified as crucial pathogenic mechanisms affecting bone cells, further exacerbating the disease process [9].

Recent clinical and experimental evidence supports the conclusion that excessive iron accumulation is an independent risk factor for osteoporosis [62,63,64]. Iron plays a crucial role in the proliferation and differentiation of both osteoblasts and osteoclasts, suggesting that iron overload is directly involved in the pathogenesis of osteoporosis [64,65,66]. Osteoclasts originate from monocyte/macrophage precursors in the myeloid lineage, whereas osteoblasts differentiate from bone marrow mesenchymal stem cells (BM-MSCs) [67,68]. The proliferation and activation of osteoclasts are regulated primarily by the receptor activator of nuclear factor-κB ligand (RANKL)/osteoprotegerin (OPG) pathway, with the RANKL/OPG ratio serving as a key indicator for assessing bone mass and integrity [69,70]. Moreover, studies have shown that osteoclast differentiation is strongly linked to the activation of transferrin receptor 1 (TFR1) and increased iron uptake [66,71]. Notably, an increase in the number of osteoclasts exacerbates the osteoporosis phenotype in animal models of iron overload [72]. Iron overload also severely impairs the biological activity of osteoblasts in vivo [73]. Runt-related transcription factor 2 (Runx2) is a critical osteogenic transcription factor, and alkaline phosphatase (ALP), which is secreted by osteoblasts, plays a pivotal role in bone mineralization [74,75]. In a zebrafish model, iron overload reduced the expression of key bone formation genes, such as ALP, collagen type I, and Runx2, in osteoblasts [76]. Additionally, a study by Balogh et al. demonstrated that osteogenic stimulation-induced upregulation of Runx2, and ALP was completely abolished at a concentration of 50 μmol/L iron, highlighting the dose-dependent negative impact of iron overload on osteoblast viability and differentiation [77]. Thus, iron overload impairs osteoblast viability and differentiation in a dose-dependent manner.

Oxidative stress refers to an imbalance between ROS production and antioxidant defense. Iron overload induces the Fenton reaction in which free iron reacts with H_2_O_2_ to generate ROS, particularly hydroxyl radicals (·OH), thereby contributing to oxidative damage [78]. While low ROS levels are essential for maintaining bone homeostasis, excessive ROS disrupt the balance between osteoblasts and osteoclasts [79]. In osteoblasts, ROS inhibit the BM-MSCs by decreasing the expression of key osteogenic markers, such as alkaline phosphatase (ALP) and collagen type I. ROS exhibit dual roles in osteoblast regulation: at low levels, they promote osteogenic differentiation by transiently activating mitogen-activated protein kinase (MAPK) pathways (such as extracellular-signal-regulated kinases, c-Jun N-terminal kinases, and p38). However, excessive or prolonged ROS exposure leads to sustained activation of stress-responsive MAPKs, particularly JNK and p38, which induces osteoblast apoptosis and impairs bone formation [80,81]. In osteoclasts, ROS promote receptor activator of RANKL-mediated osteoclast differentiation by activating the nuclear factor kappa-B (NF-κB) and MAPK pathways. Additionally, ROS inhibit autophagy by suppressing the Nrf2 and mTOR signaling pathways [82,83]. Therefore, dysregulated ROS levels may contribute to osteoporosis by impairing osteoblast differentiation and function while simultaneously promoting osteoclast differentiation and activity.

## 4. Regulatory Mechanisms of Ferroptosis in Osteoporosis

### 4.1. Impact of Ferroptosis on Osteoblast Differentiation and Bone Formation

Ferroptosis in osteoblasts has been linked to abnormal iron metabolism, where iron overload and oxidative stress impair osteoblast differentiation and function, which aligns with the mechanisms underlying ferroptosis. Jiang et al. demonstrated that iron overload induces ferroptosis in mouse embryonic osteoblast precursor cells (MC3T3-E1), inhibiting both the osteogenic differentiation and mineralization of these cells in vitro [84]. Excess iron generates ROS through the Fenton reaction, which in turn attacks polyunsaturated fatty acids in the cell membrane, triggering lipid peroxidation. This process produces lipid peroxides, such as 4-hydroxynonenal (4-HNE) and malondialdehyde (MDA), which directly damage osteoblast cell membranes, leading to apoptosis or necrosis [30]. Wnt signaling, a critical regulator of osteoblast progenitor cell apoptosis, promotes osteoblast differentiation in MSCs prior to differentiation [85]. Normally, Wnt/β-catenin activation promotes β-catenin nuclear translocation and upregulates osteogenic genes like Runx2 [86]. Thus, the Wnt/β-catenin signaling pathway plays an essential role in osteoblast differentiation and bone formation. Similarly, the bone morphogenetic protein (BMP) pathway is indispensable for regulating osteoblast differentiation [87,88]. Activation of the transforming growth factor-beta (TGF-β)/BMP pathway induces the expression of the target gene Runx2 via Smad proteins. TGF-β/BMP signaling cooperates with Wnt/β-catenin signaling to promote the osteoblastic differentiation of human mesenchymal stem cells (hMSCs) [89]. However, under ferroptotic conditions, the overproduction of ROS and lipid peroxidation damage osteoblast membranes, leading to the downregulation of Wnt signaling. This disruption impairs osteoblast differentiation, reduces bone matrix production, and ultimately decreases bone mineral density (BMD), increasing the risk of osteoporosis [90] (Figure 2). Nrf2 is a key regulatory factor that protects against oxidative and chemical damage and is expressed in various cell types, including osteoblasts, osteocytes, and osteoclasts [91]. The Nrf2-antioxidant response element (ARE) pathway has emerged as one of the primary cellular defense mechanisms against oxidative and exogenous stress [92]. During ferroptosis, excessive ROS cause Nrf2 to dissociate from its inhibitor KEAP1. Nrf2 then translocates to the nucleus, where it binds to MAF proteins and activates the ARE, initiating cellular defense mechanisms by upregulating the expression of antioxidant enzymes such as GPX4 and heme oxygenase-1 (HO-1). This process helps mitigate ROS accumulation and cellular damage [93]. Xu et al. reported that the vitamin D receptor activator 1,25-dihydroxyvitamin D3 could reduce osteoblast ferroptosis by activating the vitamin D receptor and its downstream Nrf2/GPX4 signaling pathway, which downregulates lipid peroxidation [94]. Furthermore, Ma et al. demonstrated that high glucose-induced ferroptosis in MC3T3-E1 cells (25.5 mM) could be inhibited by the endogenous antioxidant melatonin, which activates the Nrf2/HO-1 signaling pathway, reducing the levels of lipid peroxidation (LPO) and ROS while increasing GPX4 and SLC7A11 activity [95]. Thus, inhibition of the Nrf2 pathway exacerbates lipid peroxidation and further damages osteoblasts, impairing cell function and promoting cell death, ultimately inhibiting bone formation and contributing to the progression of osteoporosis. The NADPH oxidase (NOX) family serves as a significant source of ROS in eukaryotic cells, with studies showing that NOX1-5 are upregulated in Ras mutant tumors [96,97]. The IRE-like sequence in the NOX4 promoter is typically occupied by the negative transcriptional regulator IRP1. Zhang et al. demonstrated that elevated iron levels in osteoblasts induce the dissociation of IRP1, which subsequently drives NOX4 transcription. Furthermore, NOX4 knockdown reduces lipid peroxidation levels and prevents ferroptosis in osteoblasts [98]. The pathogenic mechanism underlying osteoporosis caused by ferroptosis involves the dysregulation of several key signaling pathways in osteoblasts, ultimately disrupting bone metabolic homeostasis and contributing to the development of osteoporosis.

### 4.2. Impact of Ferroptosis on Osteoclast Function and Bone Resorption

The pathogenic mechanism of ferroptosis in osteoclasts has been shown to be closely associated with the development of osteoporosis. Osteoclasts, which are multinucleated cells responsible for bone resorption, play a critical role in maintaining bone homeostasis, and their activity is tightly regulated. Ferroptosis in osteoclasts is linked to abnormalities in iron metabolism, and under conditions of iron overload, increased ROS within osteoclasts directly induce cell death. This process further enhances osteoclast differentiation and exacerbates bone resorption. The RANK/RANKL/OPG signaling pathway is central to regulating osteoclast differentiation and activation [12]. In 2009, Ishii et al. demonstrated that developing osteoclasts exhibit increased iron requirements, accompanied by elevated transferrin receptor 1 (TFR1) expression through posttranscriptional regulation. TFR1-mediated iron uptake promotes osteoclast differentiation and bone resorption activity [66]. Ferroptosis can be induced through an iron starvation response characterized by increased TFR1 expression and decreased ferritin, coupled with RANKL stimulation [99]. Inflammation plays a synergistic role in osteoclast activation. Tumor necrosis factor-alpha (TNF-α), a potent proinflammatory cytokine, activates the NF-κB signaling pathway. In the presence of TNF-α, the NF-κB pathway enhances osteoclast production and strengthens the interaction between RANKL and RANK, thereby activating downstream signaling pathways. Moreover, TNF-α acts synergistically with RANKL to directly promote osteoclastogenesis [100]. RANKL–RANK interaction activates multiple intracellular signaling cascades, including the mitogen-activated protein kinase (MAPK) pathways [98]. ROS accumulation is crucial in this process as it facilitates activation of both MAPK and NF-κB pathways, thereby increasing osteoclast differentiation and bone resorption [99,100]. Consequently, ferroptosis may contribute to osteoclastogenesis via ROS-dependent mechanisms and RANKL-mediated signaling (Figure 2). Nrf2 signaling also plays a role in the regulation of osteoclast formation and activity [91]. Knockdown of Nrf2 has been shown to induce oxidative stress and promote osteoclast differentiation under normal conditions. However, under conditions of iron overload, Nrf2 knockdown induces ferroptosis, and alleviating iron overload-induced bone loss by inhibiting osteoclast differentiation represents a potential therapeutic approach for osteoporosis [101]. In conclusion, ferroptosis influences osteoclast function by modulating several key signaling pathways, thereby increasing the risk of osteoporosis.

### 4.3. Summary of Regulatory Mechanisms

In summary, ferroptosis affects both osteoblasts and osteoclasts by disrupting iron metabolism and redox homeostasis, leading to altered differentiation and activity of bone cells. This ultimately contributes to the imbalance between bone formation and resorption observed in osteoporosis. The major signaling pathways and molecular mechanisms through which ferroptosis regulates osteoblast and osteoclast function are summarized in Table 1.

Ferroptosis contributes to osteoporosis by disrupting the balance between osteoblasts and osteoclasts. In osteoblasts, ferroptosis leads to increased ROS and MDA levels while reducing ALP and RUNX2 levels, thereby inhibiting osteoblast differentiation. In osteoclasts, ferroptosis increases ROS and MDA levels while increasing the levels of receptor activators of RANKL and MAPK, promoting osteoclast differentiation. These changes result in bone loss and osteoporosis development. Figure 2 illustrates the role of ferroptosis in both osteoblasts and osteoclasts.

## 5. Factors Affecting the Regulation of Ferroptosis in Osteoporosis

### 5.1. Factors Associated with Ferroptosis

First, ferroptosis, a form of programmed cell death, is influenced by a variety of factors, including iron metabolism, amino acid metabolism, and lipid metabolism. Iron plays a critical role in ferroptosis, with transferrin and its receptor working in concert to transport iron into cells, thereby promoting ferroptosis [110]. Amino acids, particularly cystine, are essential for this process. GSH, a key antioxidant, is synthesized through a series of enzymatic reactions that rely on the system Xc^−^ transporter to facilitate cystine uptake. As intracellular GSH synthesis is dependent on system Xc^−^-mediated cystine transport, the inhibition of system Xc^−^ results in decreased cystine uptake and reduced GSH levels, thereby promoting ferroptosis [111]. Additionally, the PUFAs is a hallmark of ferroptosis as PUFAs buildup can trigger lipid peroxidation, further driving the progression of ferroptosis [112].

Second, several drugs can modulate or influence the ferroptosis process. Erastin, a well-known ferroptosis inducer, inactivates GPX4 by inhibiting the system Xc^−^ transporter, thereby lowering intracellular GSH levels, which triggers lipid peroxidation and ferroptosis [113]. Similarly, RSL3, another ferroptosis inducer, directly inhibits GPX4 activity, promoting lipid peroxidation and ferroptosis [114]. On the other hand, ferrostatin-1 (Fer-1) serves as a ferroptosis inhibitor by scavenging free radicals and preventing lipid peroxidation, thus protecting cells from ferroptosis [115]. In addition, recent research has identified a range of other compounds that either inhibit ferroptosis, expanding the potential for therapeutic manipulation of this process. Representative agents—such as deferoxamine, melatonin, and mitochondria-targeted compounds—along with their mechanisms of action and target cell types, are summarized in Table 2.

Finally, environmental factors also play critical roles in ferroptosis. Under both physiological and pathological conditions, cellular hypoxia activates transcriptional mechanisms to regulate the expression of several genes in response to altered oxygen levels, with hypoxia-inducible factors (HIFs) serving as key regulators of the hypoxic response [116]. Under hypoxic conditions, HIFs can exacerbate certain forms of cell death. For example, in retinal epithelial cells, HIF exacerbates sodium-iodate-induced ferroptosis by upregulating superoxide dismutase (SOD), which exerts pro-oxidant effects, and by increasing the expression of the iron importers ZIP8 and ZIP14, thereby increasing iron uptake and promoting ferroptosis [117]. HIF-1α has been implicated in reducing ferritin phagocytosis and autophagic flux under hypoxic conditions. Moreover, the HIF-1α-specific inhibitor 2ME2 has been shown to prevent bone loss in ovariectomized mouse models of osteoporosis [99]. Under hypoxic conditions, HIF-2α has also been demonstrated to promote ferroptosis in tumor cells. For example, HIF-2α upregulates genes involved in lipid and iron metabolism in mouse colorectal cancer cells and colon tumors, increasing the susceptibility of these cells to dimethyl-fumarate-induced ferroptosis [118]. These findings suggest that hypoxia may precipitate alterations in cellular metabolism, particularly iron and lipid metabolism, which increases the likelihood of ferroptosis. Furthermore, chronic exposure to pollutants, such as the heavy metal cadmium, disrupts iron metabolism and peroxidative signaling pathways, triggering ferroptosis in adult mouse spermatogonia [119]. Similarly, exposure to fine particulate matter has been shown to disrupt iron metabolism and redox pathways, leading to ferroptosis in human endothelial cells [119,120]. These studies indicate that specific pollutants can exacerbate ferroptosis by increasing intracellular iron accumulation and oxidative stress, further underscoring the complex interplay between environmental factors and cellular ferroptosis.

### 5.2. Genetic Factors and Ferroptosis in Osteoporosis

The role of ferroptosis in the pathogenesis of osteoporosis has garnered increasing attention in recent years, with a concurrent rise in research examining the potential influence of genetic factors. Ferroptosis development is closely linked to disruptions in iron metabolism, highlighting the importance of iron regulatory genes in the etiology of osteoporosis. Mutations in the HFE gene, which cause hemochromatosis, result in systemic iron overload. Studies have demonstrated that mutations in HFE are associated with an increased risk of osteoporosis [121]. Iron overload due to these mutations may impact bone health through ferroptosis mechanisms.

The *SLC40A1* gene encodes FPN, a protein crucial for iron export. Deficiencies in FPN lead to excessive iron accumulation in osteoblasts, promoting osteoclastogenesis and consequently reducing bone mass [122,123]. Moreover, *SLC40A1* has been identified as a key gene involved in ferroptosis in osteoporosis [124]. Other significant genes include *TFRC*, which encodesTFR1, and *FTH1*, which encodes FTH1. Both genes play roles in iron regulation and are implicated in ferroptosis, with potential effects on osteoporosis development. Antioxidant-enzyme-encoding genes also contribute to ferroptosis regulation. The *GPX4* gene, encoding an enzyme responsible for scavenging lipid peroxides, is central in controlling ferroptosis. In patients with type II diabetes and osteoporosis, reduced GPX4 expression has been observed, suggesting increased susceptibility of bone cells to ferroptosis [125]. These findings highlight the pivotal role of GPX4 in osteoporosis pathogenesis. In summary, genetic factors affecting ferroptosis, particularly those related to iron metabolism and antioxidant defense mechanisms, are critical for understanding osteoporosis development. Further research in this area may unveil novel therapeutic targets for the management of osteoporosis.

## 6. The Therapeutic Potential of Ferroptosis in Osteoporosis

As research into ferroptosis and osteoporosis has advanced, considerable attention has been directed toward identifying potential therapeutic targets for osteoporosis, laying the groundwork for the development of novel pharmaceutical agents and facilitating the translation of fundamental research into clinical applications. Ferroptosis plays a significant role in the pathogenesis and progression of osteoporosis. Excessive iron accumulation, coupled with the accumulation of lipid peroxides, can trigger ferroptosis in bone cells. Therefore, pharmacological inhibition of ferroptosis—via iron chelation, antioxidant therapy, or specific small-molecule inhibitors—represents a promising therapeutic avenue.

### 6.1. Iron Chelators

Deferoxamine (DFO), a well-known iron chelator, has been shown to attenuate osteoporosis by modulating the expression of key ferroptosis-related proteins such as GPX4, HMOX1, and SLC7A11 in osteoblasts [84,98]. In addition to DFO, other clinically approved iron chelators, including deferiprone (DFP) and deferasirox (DFX), have demonstrated efficacy in mitigating bone loss by reducing systemic iron overload, thereby alleviating oxidative stress and inflammation associated with osteoporosis [126,127]. Further research is needed to elucidate their precise mechanisms in regulating ferroptosis and their therapeutic implications in osteoporosis treatment. Iron chelators not only reduce systemic iron accumulation but also directly inhibit osteoclast differentiation and stimulate osteoblast differentiation, demonstrating protective effects in animal models of osteoporosis. However, the clinical application of iron chelators is limited by significant limitations, including adverse effects and suboptimal patient adherence [128].

### 6.2. Natural Compounds

Natural antioxidants offer a safer alternative for modulating ferroptosis. For example, asparagine, a traditional medicinal compound, has antioxidant properties that activate the NRF2/HO-1 pathway, increase the expression of the osteogenic transcription factor Runx2, and inhibit osteoblast ferroptosis [129]. Similarly, neferine, an alkaloid derived from lotus root, possesses potent antioxidant activity, effectively suppressing osteoclastogenesis and mitigating bone loss by inhibiting RANKL-induced activation of the NF-κB signaling pathway, thus offering a potential therapeutic approach for osteoporosis [130,131]. Melatonin has demonstrated therapeutic potential in conditions of iron overload due to its strong iron-chelating capacity and potent free radical scavenging properties [128]. It has been shown to upregulate the expression of osteogenic genes, including ALP, BMP2, BMP6, osteocalcin (OCN), and OPG, primarily through activation of melatonin receptor 2 (MT2) [132]. Moreover, melatonin can also promote the osteogenic differentiation of BM-MSCs by regulating the NF-κB signaling pathway [133]. Therefore, it may be a potential treatment for osteoporosis. In addition to the compounds discussed above, Li et al. summarized multiple traditional Chinese herbal medicines such as icariin, curcumin, and quercetin, which have been shown to modulate ferroptosis via pathways related to oxidative stress, inflammation, and iron metabolism, thereby contributing to the prevention and treatment of osteoporosis [134].

### 6.3. Mitochondria-Related Targets and Therapeutic Strategies

Mitochondrial ferritin (FtMt), a mitochondria-specific iron storage protein, has emerged as a crucial regulator of ferroptosis in osteoblasts. Studies have demonstrated that FtMt suppresses osteoblast ferroptosis by mitigating oxidative stress induced by excess ferrous ions [135]. Under high-glucose conditions, overexpression of FtMt has been shown to alleviate ferroptosis in osteoblasts, while FtMt silencing promotes mitochondrial autophagy via activation of the ROS/PINK1/Parkin pathway [135]. This finding highlight FtMt as a mitochondria-associated regulatory factor and suggest its potential as a therapeutic target for the treatment of osteoporosis. In addition to intrinsic mitochondrial regulators such as FtMt, exogenous agents that target mitochondrial oxidative stress have also shown potential in alleviating ferroptosis. One such compound is Mitoquinone (MitoQ), a mitochondria-targeted antioxidant that selectively accumulates within mitochondria. MitoQ, a mitochondria-targeted antioxidant, has been shown to effectively scavenge mitochondrial ROS, preserve mitochondrial membrane potential, and reduce lipid peroxidation in various oxidative-stress-related diseases [136]. Although its direct therapeutic impact on osteoporosis has yet to be fully elucidated, recent studies have reported that MitoQ inhibits ferroptosis in chondrocytes via activation of the NRF2–Parkin signaling axis, thereby demonstrating potential therapeutic relevance [137]. These findings suggest that MitoQ and other mitochondria-targeted antioxidants warrant further investigation as potential interventions for ferroptosis-associated bone disorders, including osteoporosis.

Although many of these agents remain in the preclinical stage, their efficacy in modulating ferroptosis highlights promising therapeutic potential. The currently identified regulators of ferroptosis in the context of bone metabolism are summarized in Table 2 Further clinical studies are necessary to validate their translational value in the treatment of osteoporosis.

**Table 2 biology-14-01062-t002:** Ferroptosis-targeted therapeutic strategies and their cellular effects in osteoporosis.

Therapeutic Strategy	Mechanism	Target Cell Type	References
Deferoxamine (DFO)	Chelates Fe^2+^; modulates GPX4, HMOX1, and SLC7A11 to suppress ferroptosis	Osteoblasts	[84,98]
Asparagine	Activates NRF2/HO-1 pathway; upregulates Runx2 to inhibit ferroptosis	Osteoblasts	[129]
Neferine	Suppresses RANKL-induced NF-κB signaling to reduce osteoclastogenesis	Osteoclasts	[130,131]
Melatonin	Chelates iron and scavenges ROS; activates MT2 and NF-κB pathways to promote osteogenesis	Osteoblasts, BM-MSCs	[128,132,133]
Overexpression of FtMt	Reducing excess ferrous ions inhibits the occurrence of ferroptosis	Osteoblasts	[135]

## 7. Challenges and Future Prospects

Although increasing evidence supports a critical role for ferroptosis in osteoporosis, several significant challenges must be addressed before ferroptosis-targeted strategies can be translated into clinical therapies. First, the precise molecular mechanisms regulating ferroptosis in different bone cell populations—particularly the interactions between iron metabolism, lipid peroxidation, and cellular signaling—remain incompletely understood. Clarifying these mechanisms is essential to identify stage- and cell-specific intervention points in the context of bone remodeling.

Second, while numerous ferroptosis modulators—such as iron chelators and lipid peroxidation inhibitors—have demonstrated efficacy in vitro and in animal models, their pharmacological properties in humans—such as bioavailability, specificity, and safety—require further investigation through rigorous clinical studies.

Third, the potential crosstalk between ferroptosis and immune-related or inflammatory pathways in the skeletal microenvironment has not been adequately explored. Given the intimate connection between immune regulation and bone metabolism, understanding this interplay may uncover additional targets or synergistic strategies for intervention.

Moreover, ferroptosis appears to be involved in the pathogenesis of other bone-related diseases beyond osteoporosis. In osteoarthritis, chondrocyte degeneration has been associated with iron overload and oxidative stress, contributing to impaired cartilage homeostasis and joint dysfunction. Targeting ferroptosis in this context may offer a new therapeutic avenue to preserve cartilage integrity. In osteosarcoma, ferroptosis inducers have shown potential in preclinical studies by promoting iron-dependent cell death and lipid peroxidation in tumor cells, highlighting their promise as adjunctive anti-tumor agents.

In addition, traditional Chinese medicine has been preliminarily reported to exert anti-osteoporotic effects through ferroptosis-regulating mechanisms. Although the underlying pathways and active compounds remain to be elucidated, these findings provide an alternative perspective warranting further study.

Future research should focus on the development of specific ferroptosis modulators tailored to distinct skeletal cell types, the identification of reliable biomarkers for ferroptotic activity in bone tissue, and the systematic evaluation of therapeutic efficacy in well-controlled clinical trials. A deeper understanding of ferroptosis in various musculoskeletal disorders may ultimately enable the design of personalized, mechanism-based interventions targeting redox imbalance and iron dysregulation.

## 8. Conclusions

Ferroptosis, a regulated form of cell death characterized by iron-dependent lipid peroxidation, has emerged as a significant contributor to the development and progression of osteoporosis. Disruptions in iron metabolism, oxidative stress, and antioxidant defense mechanisms have been shown to affect bone remodeling processes by altering the function and viability of both osteoblasts and osteoclasts. A growing body of evidence implicates ferroptosis-related signaling pathways—including Wnt/β-catenin, TGF-β/BMP, TNF-α/NF-κB, MAPK, and Nrf2—in the regulation of skeletal cell fate and bone homeostasis.

Strategies aimed at regulating iron homeostasis and mitigating lipid peroxidation may help restore redox balance and preserve bone integrity. Furthermore, the mechanistic links between ferroptosis and other skeletal pathologies, such as osteoarthritis and osteosarcoma, underscore the broader relevance of ferroptosis research within the musculoskeletal field. Although substantial progress has been made in understanding the biological role of ferroptosis in bone metabolism, key challenges remain. These include elucidating cell-type-specific regulatory mechanisms, improving the translational applicability of ferroptosis-targeted agents, and evaluating their clinical efficacy and safety. This review provides a theoretical foundation for advancing ferroptosis-based therapeutic approaches in osteoporosis and highlights new directions for future investigation in musculoskeletal diseases.

## Figures and Tables

**Figure 1 biology-14-01062-f001:**
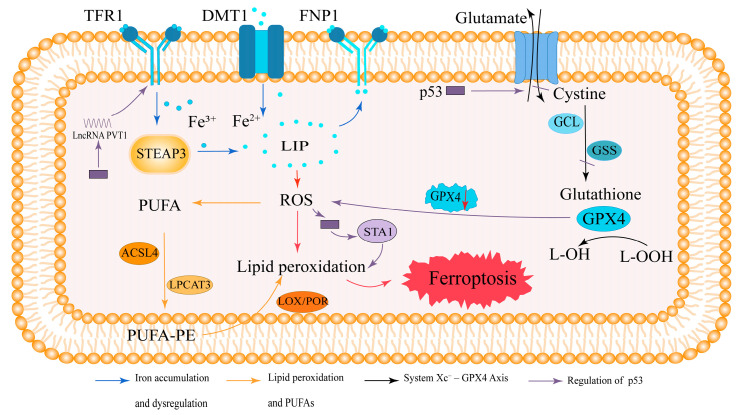
Mechanism of ferroptosis.

**Figure 2 biology-14-01062-f002:**
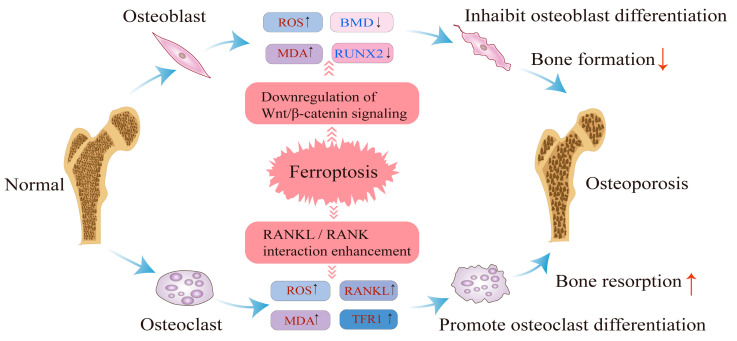
Ferroptosis occurs in osteoblasts and osteoclasts.

**Table 1 biology-14-01062-t001:** Summary of ferroptosis-related signaling pathways in osteoblasts and osteoclasts.

Signaling Pathways	Osteoblast Function	Osteoclast Function	Ferroptosis Mechanism	Reference
Wnt/β-catenin	Inhibits differentiation and mineralization	Not directly involved	Ferroptosis inhibits β-catenin nuclear translocation, reducing osteogenic activity	[102]
TGF-β/BMP	Inhibits osteogenic differentiation	No clear association	Ferroptosis downregulates BMP signaling, reducing bone matrix synthesis	[103]
Nrf2 pathway	Promote differentiation and proliferation	Inhibits production and activity	Ferroptosis suppresses Nrf2 pathway, exacerbating oxidative stress damage	[104,105,106]
NF-κB pathway	Not directly involved	Enhances osteoclast production	ROS activates NF-κB in osteoclasts	[107,108]
MAPK pathway	Not directly involved	Promotes osteoclast differentiation and bone resorption	ROS activates p38/JNK to promote osteoclastogenesis	[107,108]
PI3K/Akt/mTOR	Promotes survival and differentiation	Promotes survival and differentiation	Iron overload inhibits PI3K/Akt/mTOR in osteoblasts, reducing survival and differentiation; in osteoclasts, it activates PI3K/Akt/mTOR, enhancing survival and bone resorption	[108,109]

## Data Availability

Not applicable. This is a review article and does not contain any original data.

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
