# Peer review of "Targeting Ferroptosis: Emerging Insights into Osteoporosis Mechanisms"

_biology, 2025, doi:10.3390/biology14081062_

Round 1

Reviewer 1 Report

Comments and Suggestions for Authors

This review paper addresses the current knowledge regarding the role of ferroptosis in osteoporosis. Ferroptosis is an iron-dependent form of cell death that is characterised by an accumulation of lipid peroxidases in the cell membrane. The authors discuss how ferroptotic conditions can regulate both osteoblast and osteoclast signalling. Ferroptosis may lead to ROS generation, which stimulates osteoclasts while inhibiting osteoblasts, thereby promoting bone degradation and leading to osteoporosis. The authors further discuss how different factors such as drugs, environment, genetics, etc. can influence ferroptosis in osteoporosis. Lastly, the authors discuss the therapeutic potential of ferroptosis and how certain drugs that target ferroptotic pathways may have potential as osteoporosis treatments. The paper is very well structured and very well written. It aids the reader to easily understand the importance of ferroptosis in osteoporosis. Below are some additional comments on the paper.

- Is the review clear, comprehensive and of relevance to the field? Is a gap in knowledge identified?

The review is clear and concise. It is very well-written and easy to read and digest. The structure of the review is also excellent, and the authors should be commended on a very well-written paper.

- Was a similar review published recently and, if yes, is this current review still relevant and of interest to the scientific community?

There is a similar review from 2022. Liu P, Wang W, Li Z, Li Y, Yu X, Tu J, Zhang Z. Ferroptosis: a new regulatory mechanism in osteoporosis. Oxidative Medicine and Cellular Longevity. 2022;2022(1):2634431.

The reviewer believes that this current paper contains more novel information and a more detailed discussion on the role of ferroptosis in bone cells. However, the authors could perhaps explain the novelty of their current paper and how it differs from the previous publication.

- What specific improvements should the authors consider regarding the
methodology?

The paper is a narrative review, and therefore, no formal selection process was used for the articles included in the study.

- Are the cited references mostly recent publications (within the last 5 years) and relevant? Are any relevant citations omitted? Does it include an excessive number of self-citations?

Several relevant references have been included. Several of the references are recent publications. This demonstrates that the topic is of current relevance. A good number of references have been included.

- Are the statements and conclusions drawn coherent and supported by the listed citations?

The authors could consider elaborating more on the therapeutic potential of ferroptosis in osteoporosis. Some drugs are mentioned to show potential; however, these studies could be discussed in more detail. How do the iron-chelators manage iron-related bone loss? Why is melatonin a promising candidate? Any studies with these ferroptosis drugs in bone cells? This would help strengthen the conclusions in the paper.

Overall, the conclusions are well-written and supported by the evidence provided in the review.

- Are the figures/tables/images/schemes appropriate? Do they properly show the data? Are they easy to interpret and understand?

There are two figures and one table included in the paper. The figures are well-designed and add to the understanding of the topic. Similarly, the table is well-structured and contributes to the understanding of the topic.

Reviewer 2 Report

Comments and Suggestions for Authors

In this review article entitled “Targeting Ferroptosis: Emerging Insights into Osteoporosis Mechanisms”, Yang et al, discussed the roles and regulation of ferroptosis. Initially, the author provided an overview of ferroptosis, which then led to a discussion of its regulation. It is well reviewed. However, I have some concerns:

  1. Overview of Ferroptosis: This portion started with overview. However, it ultimately came down to the mechanism itself. It will be better for the readers if the author can put it under subheadings such as iron accumulation, lipid peroxidation etc. Ultimately, the authors have given a schematic overview in Figure 1. However, they need to mention the figure with all the mechanisms with a little more detail.
  2. Table 1: They must redo the table as what are the inducers and inhibitors of ferroptosis in the context of OB and OC.
  3. Figure 2. It needs more clarification. From the figure, no one will understand which are the inhibitors and which one are triggering the differentiation. They need to draw it carefully with a little more detail. They need to cite the figure in the text whenever it is needed. It is also required for Figure 1.
  4. “In addition, recent research has identified a range of other compounds that either induce or inhibit ferroptosis, expanding the potential for therapeutic manipulation of this process.” – Missing the ref. or details.
  5. “The Therapeutic Potential of Ferroptosis in Osteoporosis” – this portion needs to be rewritten. The author should discuss the therapeutics in more detail. Thay can divide this area under many subheadings such as iron chelator, medicinal compounds, melatonin etc. They must include a table with therapeutic information and its effects. They should discuss the mitochondrial drugs.
  6. Although the author discussed ferroptosis and OC or bone remodelling, but they did not connect it with osteoarthritis or osteosarcoma.
  7. They should include the immune system in the context of osteoporosis and ferroptosis.
  8. Signaling pathways -- they must put it in a subheading format.
  9. Overall, the article is missing novelty and uniqueness. They should include challenges and future prospects or unknown facts. It requires subheadings for each heading.

Reviewer 3 Report

Comments and Suggestions for Authors

The incidence rate of osteoporosis is getting higher and higher, which has a greater impact on human health. A deep understanding of the molecular mechanism of bone loss will help to develop new targeted therapies. As ferroptosis plays an important role in many diseases, its role in osteoporosis cannot be ignored. The author of this article summarized the mechanism of ferroptosis and its role in osteoporosis, providing a reference for researchers to conduct in-depth research, which has certain academic value. Unfortunately, there have been many similar reviews, one of which has been published in biology in 2025 (Li P, Xu TY, Yu AX, Liang JL, Zhou YS, Sun HZ, Dai YL, Liu J, Yu P. The Role of Ferroptosis in Osteoporosis and Advances in Chinese Herbal Interventions. Biology (Basel). 2025 Apr 2;14(4):367. doi: 10.3390/biology14040367IF: 3.5 Q1 . PMID: 40282232; PMCID: PMC12025301). The above review provides a more comprehensive summary. In addition, there have been many studies related to ferroptosis and osteoporosis in 2025, but this article has not cited them, resulting in insufficient innovation. This is also the biggest problem with this article. There are also some minor issues, listed as follows:

  1. Figures 1 and 2 can be drawn more delicately. It is best to draw the cell nucleus in Figure 1, and the captions of the other two figures should be different from the font or font size in the main text.
  2. On line 156, there are only half parentheses.
  3. Lines 184-186 can be expressed more concisely.
  4. In acknowledgements, it is usually to thank someone or something other than the author.

Reviewer 4 Report

Comments and Suggestions for Authors

The Review «Targeting Ferroptosis: Emerging Insights into Osteoporosis Mechanisms» by Hailing Yang et al. is very interesting. 

The authors explore the role of ferroptosis in osteoporosis, providing a theoretical framework for the clinical application.

The minor aspects:

Point 1

Line 336-338 - The authors wrote «Recent research has identified a range of other compounds that either induce or inhibit ferroptosis… » They fail to provide links or, at the very least. Please, to clarify or delete these lines.

Point 2

The abbreviation is often repeated in the text, for example - reactive oxygen species (ROS)… I recommend to expand all abbreviation in special section. 

In general, abbreviations should be explained at their first mention...

Point 3

Finally, the discussion could be further expanded to address additional perspectives on osteoporosis treatment. For instance, the authors might consider discussing the potential role of traditional Chinese medicine in managing osteoporosis.

Round 2

Reviewer 2 Report

Comments and Suggestions for Authors

The review article written by Yang et al., described how ferropstosis could be a new treatment strategy for restoring the imbalance of osteoporosis. The revised manuscript has been significantly upgraded and well written. However, I have minor concerns:

  1. "Regulation of Ferroptosis by p53": although the author discussed the inhibitor of Glu-Cys transporter, they must add all the other downstream effectors of p53. such as lncRNA, p21, SAT1 enzyme (They mentioned it in PUFA section).
  2. Fig 1. Summary of Ferroptosis Mechanism: It will be helpful for the readers if the author can improve the fig by mentioning inside the fig which mechanism is regulating which pathway.

Reviewer 3 Report

Comments and Suggestions for Authors

The quality of the paper has improved significantly. I have no other comments.

Author Response

We sincerely thank the reviewer for their positive feedback and appreciation of the improvements made. We are glad that the revised manuscript meets the reviewer's expectations.